# Renaissance Distribution for Statistically Failed Experiments

**DOI:** 10.3390/ijms20133250

**Published:** 2019-07-02

**Authors:** Roman Popov, Girish Karadka Shankara, Clemens von Bojničić-Kninski, Alexander Nesterov-Mueller

**Affiliations:** Institute of Microstructure Technology, Karlsruhe Institute of Technology (KIT), 76344 Eggenstein-Leopoldshafen, Germany

**Keywords:** peptide arrays, statistics, deep substitutions

## Abstract

Much of the experimental data, especially in life sciences, is considered to be useless if it demonstrates a large standard deviation from the mean value. The Renaissance distribution, as presented in this study, allows one to extract true values from such statistical data with large noise. To obtain proof of the Renaissance distribution, high-throughput synthesis of deep substitutions for a target amino acid sequence was performed, and the known epitope was identified in assay with human serum antibodies. In addition, the Renaissance distribution was shown to approach the epitope affinity maturation by the deep alanine substitution. The Renaissance distribution may have an impact in the development of novel specific drugs.

## 1. Introduction

Experimental science is based on well-established statistical tools used to estimate the quality of the data measured. The determination of the mean value with a relatively small standard deviation is considered a successful measurement. Nevertheless, there is much experimental data, especially in life sciences, that has statistically failed but may be “interesting”. This failure is often caused by poorly defined experimental conditions or some processes that are barely understood. Such results are frequently observed, for example, in the case of novel multicomponent reactions [1], where the product yield is not optimized, or, in the case of combinatorial synthesis, where novel molecules are synthesized with unknown properties [2,3]. Frequently, enzymes demonstrate functional heterogeneity, which cannot be represented by employing classical bulk reactions and classical statistics [4]. For example, single-molecule experiments have revealed a dynamic disorder in the subsequent catalytic cycles of cholesterol oxidase [5], horseradish peroxidase [6], and lipase B [7]. From this point of view, the statistically unreliable experimental data may include valuable scientific information that can help improve the experiment or understanding of those processes. The generally accepted opinion is that the scientist has to improve his experiment to get better statistics before presenting the results to the scientific community, but this very step remains a bottleneck in comparison with the modern communication and possibilities of generating a large amount of experimental data. Dealing with their own imperfect experiments, the scientist is limited in communicating about them because he will not publish statistically failed results.

In this study, we presented a novel synthesis of substitutional peptide arrays and, using this example, proposed a novel statistical distribution to interpret the data measured. The distribution proposed in paper is called “Renaissance,” by analogy with the period in European history which is related to several Italian cities, but that would not exhibit any Renaissance features in the case where the development level over all European cities of that time is averaged. The Renaissance distribution is a novel statistical method for cases when the standard deviation from the mean value is higher than 30%. In general, the Renaissance distribution is suitable for automatically searching for artifacts and providing a reliable comparison of the signals by varying parameters of a system with a high noise.

## 2. Results and Discussion

The principle of the stochastic array synthesized in this study is shown in Figure 1. The stochastic array delivers all possible substitutions of a specific amino acid sequence with amino acid alanine. The synthesis of all substitutional sequences occurs by stochastically assembling of the amino acid particles (Figure 1a) on the cavities (Figure 1b) functionalized with amino groups. 

The shape of the cavities was chosen so that only one particle fits inside. The particle patterns were detected with a fluorescent scanner equipped with a visible light filter (Figure 1c,d). Then, amino acids were extracted from the particles and coupled to the bottom of the corresponding cavities, building peptide bonds with amino acid groups on the surface. After coupling, the particles were removed from the cavities with an ultrasound (Figure 1e). Subsequently, in the wells that did not capture particles, the amino acid alanine was coupled out of the solution. After the coupling of the amino acids from the particles and solution, the protective groups were removed so that the cavities were ready for the next coupling step. Using this principle, a stochastic alanine substitution array was synthesized for the amino acid sequence KEVPALTAVETGAT. This peptide was reported to be reactive for 90% of the European blood sera [8].

Theoretically, 2^11^ = 2048 different peptides should be generated according to all possible substitutions of the target peptide KEVPALTAVETGAT with alanine. By analyzing the particle deposition, only 1142 different peptides were displayed on the chip. For the target sequence, 2672 copies were identified. 

Figure 2 shows the distribution of the fluorescent signal intensity of the peptide spots across these copies, whereby the peptides are arranged according to the descending signal. This curve was approximated with the logarithmic function:

y(x) = −*a*·ln (x) + *b*,
(1)
where *a* = 92, *b* = 834, and the coefficient of determination is R^2^ = 0.895. Formula (1) represents a general function that is defined as the Renaissance distribution. In an ideal case, all the signals would have the same fluorescent intensity. However, because of the statistically non-ideal amino acid extraction from the particles, the majority of signals are close to the chip surface background value of approx. 200 a.u., and only few high-intensity signals are demonstrated by the first 50 peptides. Formula (1) fit the signal distributions for all substitutions of the target sequence very well, with coefficients of determination > 0.8. Interesting to note is that removal of the signals with maximal intensity significantly increased the coefficient of determination to R^2^ = 0.97. Such deleted points will be considered as artifacts.

Apparently, the estimation of the mean value for the signal intensity over all peptide copies in Figure 2 will give a false result, which is close to the background value. Instead of using mean value, the coefficient b in the Formula (1) was considered to represent the true intensity signal. In this case, the term −*a*·ln(x) is a negative entropy responsible for inefficient coupling. The crucial feature of the Renaissance distribution (1) is that the true value, parameter b, is independent of the number of the trials. If a set of x copies will be reduced, that would correspond to the set of x^β^. Thus, the parameter β will only modify the negative information entropy *a* to *a*·β. The smaller the number of trails, the higher the coefficient *a will be*. The differential presentation of Formula (1)
δy = −δx/x
(2)
reveals the meaning of the Renaissance distribution. Small changes of the feature parameter y of a system is inversely proportional to the number of the non-ideal system states. In particularly, this law means that the appearance of a small number of the non-ideal states in the system with the zero entropy has a significant influence on the feature parameters. Vice versa, if the number of non-ideal states is already large, the appearance of the additional non-ideal states will have only marginal influence on the feature parameter y. Thus, the Renaissance distribution appears both as a statistical distribution and as a law that describes the decay of a state with the zero entropy.

To prove the Renaissance distribution, we applied an algorithm to all the substitutional peptides to eliminate the copies (artifacts) until the remaining points could be fitted using the Renaissance distribution, with R^2^ > 0.95 for each specific sequence (Figure 3).

Using the Renaissance distribution (1), the coefficient *b* (true value) was estimated for the single alanine and double alanine substitutions of the target sequence to obtain the binding motif of the serum antibodies (Figure 4). The alanine substitutions revealed the binding motif **K**XXXXXX**VETG**X**T**. This result corresponds to the known epitope **LTAVETG**, which is part of the poliovirus and can also be found in the human epitope database (iedb.org) [9]. The location of the target peptide in the capsid protein VP1 of the poliovirus [10,11] is shown in the Appendix A online. By contrast, the estimation of the fluorescent signals using classical distribution, which is based on the mean values over all peptide copies, delivers significantly different results in which the fluorescent signal of the target peptide was not distinguished from the background.

The Renaissance distribution fit the fluorescent signals from different peptides very well, with a coefficient of determination of R^2^ = 0.96. Interestingly, that deep alanine substitutions cause the decay of the epitope affinity, according to the Renaissance distribution. The deep alanine substitution reveals that the difference in the fluorescence signal between epitopes and its substitutions was not more than 40%. This fact could explain the off-target activity of antibodies which are simultaneously applied for specific labeling of the proteins. From this point of view, the Renaissance distribution can be used for the development of the novel specific drugs in silico. First, to increase the affinity of the initial scaffold, novel building blocks should be tested according to the minimization of the binding energy. Then, only those building blocks should be accepted, which increases the affinity of the scaffold along the Renaissance distribution. Such an approach would mimic the natural maturation of the epitopes.

## 3. Materials and Methods

### 3.1. Synthesis of Peptide Sequences

The full-size microstructured substrates (75 mm × 25 mm × 1 mm) were made of fused silica using photolithography and reactive ion etching (RIE). The array of cylindrical microwells, with diameter of 12 μm and the depth of 9 μm, has a pitch size of 20 μm. 

The crosslinked poly(methyl methacrylate) (PMMA) microbeads (MICROBEADS®) were selected as solid carriers for the amino acid derivatives. They were manufactured by emulsification polymerization and crosslinking at 3%. The microspheres had a mean diameter of 10 μm with an extremely narrow size distribution. The coefficient of variation was declared to be less than 5%. The commercially available amino acid derivative (fluorenylmethoxycarbonylamino acid pentafluorophenyl ester) was used as monomer. The amino acid derivative was introduced into the polymer matrix of the microbeads. Crosslinked PMMA microspheres (1 g) were dispersed in 5 mL of the amino acid derivative solution in dichloromethane (DCM). The mass per volume fraction (%, (*w/v*)) of amino acid in the DCM solution was adjusted for each type of monomer. The dispersion was gently stirred until a paste-like medium was obtained. The paste underwent drying overnight at normal conditions in an open vessel in a fume hood. The resulting dry mass of microbeads was milled in a falcon tube with two metal spheres (Ø 5 mm) on a vortex shaker.

Prior to applying the amino acid particles, the functional surface underwent Fmoc (Fluorenylmethoxycarbonyl) deprotection in a solution of 20% (*v/v*) piperidine in Dimethylformamide (DMF) for 30 min at room temperature. The residues of the deprotection solution were removed by washing in DMF for 5 min twice, in methanol for 3 min twice, and in DCM for 1 min once. The slide was dried in a flow of argon. 

The deposition of the microbeads was performed in two steps. First, the dry mixture of the microbeads was spread over the surface of the microstructured substrate with a soft lint-free tissue. Although the microstructures were completely filled with the microbeads, the top surface of the substrate still contained a monolayer of excessive microbeads. This excess was removed in the second step with the flow of compressed air applied tangentially to the surface of the substrate. The air flow totally removed the excessive microbeads from the top surface, whereas the microwells remained filled. The imaging of the microbeads’ deposition pattern was performed using a fluorescence scanning with optical density filters.

The extraction of the amino acid derivatives from the microbeads took place in a saturated vapor of organic solvent inside a special chamber. The slide holders with the substrates were inserted into the chamber for 1 min each. After taking the slides out of the chamber, the process was repeated 5 times to enhance the extraction of the amino acid derivatives from the microbeads. The subsequent coupling step was performed in an oven under argon for 60 min at 90 °C. 

Removal of the microbeads from the microwells was performed in several steps by placing the substrate in a mixture of 5% (*v/v*) MEA (2-aminoethanol) in acetone and exposing it to acoustic waves in an ultrasonic bath for 2 min twice. Alanine (Fmoc-protected, OPfp (pentafluorophenyl)-activated) was coupled to the remaining free amino groups of the substrate from 0.1 M solution in DMF for 60 min at room temperature.

The remaining free amino groups were capped in a solution of 10% (*v/v*) acetic anhydrate and 20% (*v/v*) DIPEA (N,N-Diisopropylethylamine) in DMF for 10 min at room temperature repeated twice. The residues of the capping solution were removed by washing in DMF for 5 min twice, in methanol for 3 min twice, and in DCM for 1 min once. The slide was dried in a flow of argon.

The steps of Fmoc deprotection, amino acid coupling, and capping were performed multiple times until the full-length peptides were synthesized.

### 3.2. Immunostaining

The array was incubated with a blood serum from a healthy donor diluted 1:1000 in PBS-T (Phosphate Buffered Saline with Tween) overnight at 4 °C and then stained with the secondary anti-human-IgG antibodies. The fluorescent signals from the spots stained with secondary antibodies were read out with a fluorescent scanner.

### 3.3. Statistical Analysis

The general function of the Renaissance distribution is presented by a logarithmic formula (1). To fit this formula, a Python program (Python 3.7) was used. The Python codes are available as a separate file in the Appendix A online. The initial data included approximately 78,000 peptides and their corresponding immunostaining signals. The program (a) separates unique sequences; (b) generates the Renaissance distribution for each unique sequence and calculates coefficients a (negative entropy) and *b* (true signals) with a defined minimum coefficient of determination R*^2^_min_*. The increasing of the initial R*^2^* (if R*^2^* < R*^2^_min_*) is realized by searching for single peptide copies that are responsible for the low R*^2^* and their deletion. The program also shows how many copies were deleted to reach R*^2^_min_* and the resulting R*^2^*. Python is an open source programming language and was download at https://www.anaconda.com/distribution/.

In the scope of this program, a common method of logarithmic fitting was used: all initial signals x*_i_* were replaced with X*_i_* = ln(x*_i_*) and a linear regression with the (X*_i_*,y*_i_*) was performed to get a logarithmic fitting. An additional simple example of fitting experimental data with a large standard deviation using the Renaissance distribution is demonstrated in the Appendix A online. All data processing (calculation of mean value, standard deviation, and logarithmic approximation) was performed using an Excel program (Appendix A).

## 4. Conclusions

The stochastic high-throughput synthesis of deep substitutions for a target amino acid sequence was presented. Deep substitution peptide chips were used for studying interactions with antibodies in a high-density-array format. Renaissance distribution was developed to analyze the fluorescent signals from the peptide–serum antibody interactions. Renaissance distribution differs substantially from the statistics based on calculations of the mean values. This allows for identification of the artifact signals which deviate from the Renaissance distribution. In contrast to the other statistical distributions, the Renaissance distribution represents the law of the decay of states with zero entropy. Its application was demonstrated for the known polio epitope. 

The Renaissance distribution may have an impact in life sciences and organic chemistry for the analysis of experimental data produced under poorly defined stochastic conditions with a large number of trials. The example with deep substitutions for the target epitope demonstrates the principle of Renaissance increasing of the binding affinity, which can be used in the development of novel specific drugs.

## Figures and Tables

**Figure 1 ijms-20-03250-f001:**
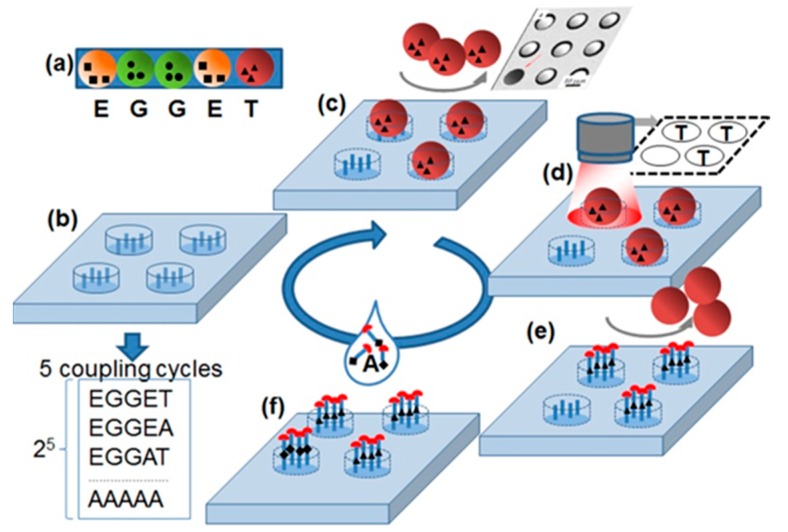
Stochastic substitutional arrays with amino acids particles. (**a**) Each amino acid particle carries a special amino acid with protecting groups. (**b**) Functionalized substrate is exposed to (**c**) one kind amino acid particles. The particles fill in the cavities stochastically. (**d**) The particle pattern is detected with a camera. (**e**) The amino acids are extracted from the particles and coupled inside the cavities via peptide bonds. The particles are removed from cavities. (**f**) The slide is exposed to a solution containing the amino acid alanine. After the removal of protecting groups (red half circles), a new cycle can be started. The repetition of 5 cycles results in the synthesis of peptides with all possible Ala substitutions in original sequences TEGGE.

**Figure 2 ijms-20-03250-f002:**
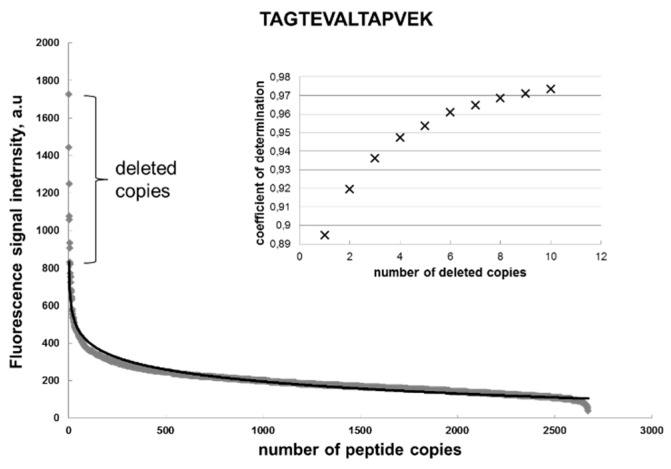
Distribution of the fluorescent signal intensity of the identical peptides (gray diamonds) and its logarithmic approximation (black) on a chip. The curves represent the Renaissance distribution. y = −92ln(x) + 834; R^2^ = 0.895. The inlet shows an increase of the R^2^ by deleting the copies from the original distribution of the fluorescent intensity.

**Figure 3 ijms-20-03250-f003:**
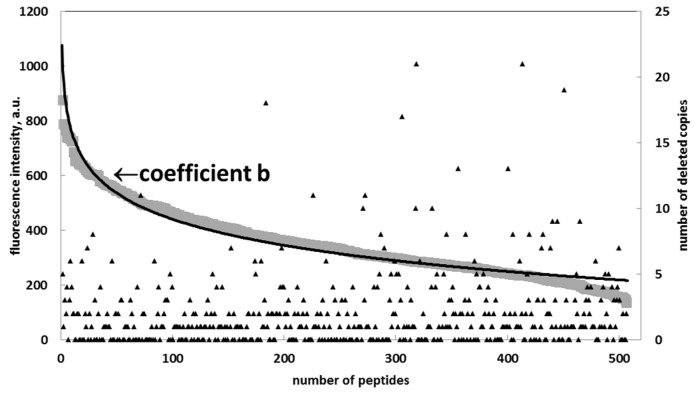
Distribution of coefficient b for peptides representing multiple alanine substitutions of the target sequence TAGTEVATLAPVEK (diamonds) and its approximation (black) with a Renaissance distribution: a = −133, b = 1051, and R^2^ = 0.96.

**Figure 4 ijms-20-03250-f004:**
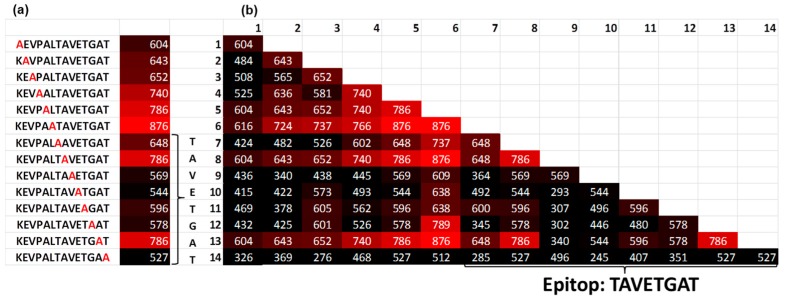
(**a**) Single and (**b**) double Ala substitutions of the target peptide KEVPALTAVETGAT. The number indicates the fluorescence intensity. In the table (**b**), the position (x,y) means that Ala was placed in position x and position y in the target peptide. The intensity of the signals reveals a binding motif **KXXXXXXVETGXT** (**c**), indicating the location of the target peptide in the capsid protein VP1 of the poliovirus.

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
