# Peer review of "Renaissance Distribution for Statistically Failed Experiments"

_ijms, 2019, doi:10.3390/ijms20133250_

Round 1
Reviewer 1 Report
The manuscript entitled " Renaissance distribution for statistically failed experiments" by Dr. Alexander Nesterov-Mueller et al, recommended for publication after the following comments are addressed.
v I suggest to include the conditions used for synthesis of peptide sequences, amino acid concentration, deprotection conditions, ect.
v I suggest to show location of the target peptide in the capsid protein in bigger size.
Author Response
I suggest to include the conditions used for synthesis of peptide sequences, amino acid concentration, deprotection conditions, etc.
To describe the synthesis of peptide sequences in more details, we have divided the section 2 in four subsections: 2.1 Principle of stochastic substitutional arrays, 2.2 Synthesis of peptide sequences, 2.3 Immunostaining and 2.4 Statistical analysis. Hereby, in 2.2 we have described more detailed the solid phase chemistry steps used in this work.
I suggest to show location of the target peptide in the capsid protein in bigger size.
We have introduced as supplementary Figure S5 to show location of the target peptide in bigger size. The small image in Figure 4c has been removed.

Reviewer 2 Report
The authors tried to use a so-called Renaissance distribution to “rescue” statistically failed experiments. Overall, this is a pretty interesting topic. I totally agree that many “non-significant” experimental data are due to improper analysis or under analysis. However, this study didn’t provide enough evidence to show the Renaissance can really work. The conception of ‘Renaissance’ is not even clear. The authors need to give a clear definition to it and give the limitation of the usage.
1. How the data were fitted? With what’s algorism?
2. What is the “renaissance curve” and “renaissance distribution”? The authors need to provide more details of it, such as a general function.
3. Can the renaissance distribution only be applied to logarithmic distributed data? How about other type of data?
4. At least, the authors need to use some simulated data to show there is a general application of Renaissance analysis for different type of data and find its limitation.
Author Response
1. How the data were fitted? With what’s algorism?
We have provided (Supplementary Materials online) a Python program (Python 3.7), which was used to fit the experimental data. The initial data included approximately 78000 peptides and their corresponding immunostaining signals. The program (1) separates unique sequences; (2) generate for each unique sequence Renaissance distribution and calculate coefficients a (negative entropy), b (truth values) with defined minimum coefficient of determination R2min. The increasing of initial R2 (if R2 < R2min) is realized by searching for single peptide copies that are responsible for low R2 and their deletion. The program also shows how many copies were deleted to reach R2min.
Python is open source programming language and could be download under: https://www.anaconda.com/distribution/
In the scope of this program, the popular method of logarithmic fitting was used: all xi were replaced with Xi = ln(xi) and then a usual linear regression with the (Xi,yi) will be done.
To describe this, we have introduced additional section 2.4.
2. What is the “renaissance curve” and “renaissance distribution”? The authors need to provide more details of it, such as a general function.
The “renaissance curve” and “renaissance distribution” were used in the original text as synonyms. To omit an ambiguity, we have replaced the “curve” with “distribution”.
We have inserted the definition of the Renaissance distribution as a general function both in the section 2.4 and in the text next to formula (1). It is actually the formula (1).
3. Can the renaissance distribution only be applied to logarithmic distributed data? How about other type of data?
According to our opinion, the Renaissance distribution has only the logarithmic form. This form is the broadly used in the statistical physics to describe the influence of the growing number of possible microscopic configurations on macroscopic parameters of a system. To do this, the definition of entropy is used: S = k log (). In our case, the term –aln (x) represents growing number of possible “distortions” of the true value b.
We underlined in the text, that logarithmic form originated from the law: y - x/x. This means that the decay of the true signal is proportional to the number of possible microscopic configurations. Integration of this formula gives y = - ln (x).
Theoretically, one can introduce general “decay” law y - x*f(x), where f(x) is an arbitrary function related to the number of system states. Maybe, such f(x) exists for strongly nonlinear systems. But according to our experimental experience, the logarithmic dependence works very well to interpret the signals from molecular interactions in the array format.
4. At least, the authors need to use some simulated data to show there is a general application of Renaissance analysis for different type of data and find its limitation.
Because some readers cannot be familiar with the Python, we have inserted a simple general example of application of Renaissance distribution using built-in functions of EXCEL program (Supplementary Materials online).
Thus, using these steps (Figures S1-S4), the Renaissance distribution and its comparison with the classical “mean value-based” statistics can performed for an arbitrary data set without special mathematics/programming skill.
We believe that Renaissance distribution may be applied if the classical statistical analysis delivers unreliable statistics: for example, if the deviation from the mean value is more than 30%. The Renaissance distribution allows for automatic finding for artifacts, while the separation of the points with strong deviation in the scope of the classical statistical analysis is an voluntary action. In general, Renaissance distribution is suitable for reliable comparison of signals by varying parameters of a system with a strong noise.
The application of the Renaissance makes no sense for obtaining a mean value in low-noise systems, where the conservative (i.e. representable with potential functions) forces are dominated. The renaissance distribution cannot be applied if number of experiments is “small”. For example, 4 signals can be always approximated with different functions with a relative high R2, so the law of the entropy decay y - x/x cannot be clearly distinguished

Reviewer 3 Report
The study is interesting for most of the researchers. The authors could include a few more points to justify the findings in the Introduction. Though, there are numerous studies are reported, this justification will lead to a clear understanding of data representation.
Information about the statistical analysis should be included in the methods section.
In Figure 1, authors specified two b) legends it makes unclear for the readers. It can be combined at the beginning of the sentence.
There is no information about the capsid protein VP1 structural details.
Author Response
The study is interesting for most of the researchers. The authors could include a few more points to justify the findings in the Introduction. Though, there are numerous studies are reported, this justification will lead to a clear understanding of data representation.
The Renaissance distribution is a statistical method for the cases when the standard deviation from the mean value is more than 30%. In general, Renaissance distribution is suitable for automatic search for artifacts and the reliable comparison of the signals by varying parameters of systems with high noise.
We have introduce these justifications into the Introduction.
Information about the statistical analysis should be included in the methods section.
We have provided (Supplementary Materials online) a Python program (Python 3.7) which was used to fit the experimental data. The initial data included approximately 78000 peptides and their corresponding immunostaining signals. The program (1) separates unique sequences; (2) generate for each unique sequence Renaissance distribution and calculate coefficients a (negative entropy), b (true values) with defined minimum coefficient of determination R2min. The increasing of initial R2 (if R2 < R2min) is realized by searching for single peptide copies that are responsible for low R2 and their deletion. The program also shows how many copies were deleted to reach R2min and the resulted R2.
Python is open source programming language and could be download under: https://www.anaconda.com/distribution/
In the scope of this program, the popular method of logarithmic fitting was used: all xi were replaced with Xi=ln(xi) and then a usual linear regression with the (Xi,yi) will be done.
In addition, we demonstrated a simple general example of application of Renaissance distribution using built-in functions of EXCEL program (Supplementary Materials online).
In Figure 1, authors specified two b) legends it makes unclear for the readers. It can be combined at the beginning of the sentence.
The correction has been made to avoid ambiguity.
There is no information about the capsid protein VP1 structural details.
We have introduced supplementary Figure S5 to show location of the target peptide in bigger size. The small image in Figure 4c has been removed.
The location was obtained using VMD: Visual Molecular Dynamics (Humphrey, William; Dalke, Andrew; Schulten, Klaus (February 1996). "VMD: Visual molecular dynamics". Journal of Molecular Graphics. 14 (1): 33–38). The 3D structure of the capsid protein VP1 of the poliovirus is accessible in Protein Data Bank (http://www.rcsb.org/structure/3J8F).
Corresponding information has been inserted into the figure capture and the reference list.

Reviewer 4 Report
This study presents a very important aspect of scientific study - is arithmetic mean a good representation in scientific studies. An analogy will be meandering water flow as the path of least resistance but arithmetic mean is likened to the shortest path. Hence, the authors are correct to point out using their analogy of Italian cities. Based on this, I will recommend acceptance. However, I found several issues / inadequacy that I will like the authors to address.
On line 104, what does "Theoretically 212 = 2048 different peptides should be generated" means? It is incomprehensible to me.
I observed that comma is used as decimal point (such as in line 121 "0,895"). I will suggest to use dot as decimal point instead (eg, "0.895") as comma tends to be used as thousands separator.
The authors used the b0 coefficient in logarithmic function to represent the mean. However, since this study is to propose a new distribution, Renaissance distribution, it will require more stringent statistical treatment. The authors will have to provide the corresponding equations to calculate at least the (a) mean, (b) probability moment function, and (c) cumulative density function; from raw data. This will also be a method for readers to calculate the parameters of Renaissance distribution for their own work.
Author Response
This study presents a very important aspect of scientific study - is arithmetic mean a good representation in scientific studies. An analogy will be meandering water flow as the path of least resistance but arithmetic mean is likened to the shortest path. Hence, the authors are correct to point out using their analogy of Italian cities. Based on this, I will recommend acceptance.
However, I found several issues / inadequacy that I will like the authors to address.
On line 104, what does "Theoretically 212 = 2048 different peptides should be generated" means? It is incomprehensible to me.
Sorry for this typing error. The proper text is 211 = 2048. The correction has been made.
The length of the target peptide is 14. It has 3 Ala. So, there is only 11 positions, each of which can be substituted by Ala from the solution. This results in 2048.
I observed that comma is used as decimal point (such as in line 121 "0,895"). I will suggest to use dot as decimal point instead (eg, "0.895") as comma tends to be used as thousands separator.
The corresponding corrections have been made.
The authors used the b0 coefficient in logarithmic function to represent the mean. However, since this study is to propose a new distribution, Renaissance distribution, it will require more stringent statistical treatment. The authors will have to provide the corresponding equations to calculate at least the (a) mean, (b) probability moment function, and (c) cumulative density function; from raw data. This will also be a method for readers to calculate the parameters of Renaissance distribution for their own work.
We have inserted a simple general example of application of Renaissance distribution using built-in functions of EXCEL program (please see new Supplementary Materials). Using these steps (Figures S1-S4), the Renaissance distribution and its comparison with the classical “mean value-based” statistics can be performed for an arbitrary data set without special mathematics/programming skill.
For more automation of the data processing, we have attached a Python file (Version 3.7 is freely available online).
The mathematical apparatus of the mean-value statistics hardly can be used to describe true value (coefficient b) and correspondingly to derive for it classical statistics functions as probability moment function. The reason for that is that b is the result of the curve approximation and its uncertainty is determined by the approximation coefficient R2.
This fact defines the restrictions of the Renaissance distribution as well its advantages. The Renaissance distribution may be applied only if the classical statistical analysis delivers unreliable values: for example, if the standard deviation from the mean value is more than 30%. The Renaissance distribution allows for automatic finding for artifacts, while the separation of the points with strong deviation in the scope of the classical statistical analysis is an voluntary action. In general, Renaissance distribution is suitable for reliable comparison of signals by varying parameters of a system with a strong noise.
An interesting historical example of a noise system is the telegraph in the early stage of its application. Because the cable connection was constantly disturbed by random effects, the same word was transmitted several times till it was finally recognized on the other end of the cable. While the telegraph could be technically improved, the interaction of biomolecules is a reality which is exposed to several random effects. Therefore, if the molecular interactions are studied in array format, one tries to use as many copies of spots as possible to get reliable statistics. This noise can originate from the additional state of the target molecules which have lower affinity to the probe. In the statistical physics, the influence of the number of the system states on the interactions is described by entropy S ~ log (). But this is exactly the logarithmic term that appears in the Renaissance distribution.

Round 2
Reviewer 2 Report
The authors addressed my questions very well. I suggest to accept it.